# Urine Molecular Biomarkers for Detection and Follow-Up of Small Renal Masses

**DOI:** 10.3390/ijms232416110

**Published:** 2022-12-17

**Authors:** Algirdas Žalimas, Raimonda Kubiliūtė, Kristina Žukauskaitė, Rasa Sabaliauskaitė, Mantas Trakymas, Simona Letautienė, Edita Mišeikytė Kaubrienė, Jurgita Ušinskienė, Albertas Ulys, Sonata Jarmalaitė

**Affiliations:** 1National Cancer Institute, 08660 Vilnius, Lithuania; 2Institute of Biosciences, Life Sciences Center, Vilnius University, 10257 Vilnius, Lithuania; 3Faculty of Medicine, Vilnius University, 03101 Vilnius, Lithuania

**Keywords:** small renal mass, active surveillance, renal cell carcinoma, renal mass biopsy, prognostic markers

## Abstract

Active surveillance (AS) is the best strategy for small renal masses (SRMs) management; however, reliable methods for early detection and disease aggressiveness prediction are urgently needed. The aim of the present study was to validate DNA methylation biomarkers for non-invasive SRM detection and prognosis. The levels of methylated genes *TFAP2B*, *TAC1*, *PCDH8*, *ZNF677*, *FLRT2,* and *FBN2* were evaluated in 165 serial urine samples prospectively collected from 39 patients diagnosed with SRM, specifically renal cell carcinoma (RCC), before and during the AS via quantitative methylation-specific polymerase chain reaction. Voided urine samples from 92 asymptomatic volunteers were used as the control. Significantly higher methylated *TFAP2B*, *TAC1*, *PCDH8*, *ZNF677,* and *FLRT2* levels and/or frequencies were detected in SRM patients’ urine samples as compared to the control. The highest diagnostic power (AUC = 0.74) was observed for the four biomarkers panel with 92% sensitivity and 52% specificity. Methylated *PCDH8* level positively correlated with SRM size at diagnosis, while *TFAP2B* had the opposite effect and was related to SRM progression. To sum up, SRMs contribute significantly to the amount of methylated DNA detectable in urine, which might be used for very early RCC detection. Moreover, *PCDH8* and *TFAP2B* methylation have the potential to be prognostic biomarkers for SRMs.

## 1. Introduction

Small renal masses (SRMs) are incidentally detected kidney tumors, which are less than or equal to 4 cm in diameter and usually correspond with stage T1a renal cell carcinoma (RCC) [1]. To date in clinical practice, there are no specific tests for SRM detection, and although most patients are diagnosed with localized kidney cancer, the true incidence of SRMs is unknown [2,3]. Regardless of earlier detection and treatment, epidemiologic studies have demonstrated stable RCC mortality, reflecting possible overdiagnosis and overtreatment [3,4]. Suitable clinical management of SRMs is crucial despite their slow growth and weak metastatic potential. Surgical resection or nephrectomy remains the main treatment strategy. However, almost 20–30% of SRMs that are suspicious for malignancy by preoperative imaging are disclosed to be benign upon final pathological examination after surgery [5]. Furthermore, a greater part of incidental tumors is detected in elderly persons with comorbidities, for whom, the surgical treatment risk outweighs the low oncologic risk. A possible alternative is percutaneous ablation of the SRM, but there are still risks of complications or local recurrence [6].

The active surveillance (AS) strategy provides the ability to delay an active treatment and is the best strategy for SRMs management; its oncologic safety is reported by numerous studies [7,8,9,10,11,12]. However, there is a possibility for metastatic progression (~2% of cases) [13] and rapid tumor growth, causing delayed interventions and/or additional treatment [7]. Without histological examination, the only parameter helping to predict the aggressiveness of SRMs is their growth rate, which is observed by periodic clinical and radiological evaluation [14]. Thus far, there are no available molecular biomarkers that could assist in timely SRM detection and accurately predict the progression of SRMs. Although a recent study identified protein-based prognostic biomarkers for SRMs [15], DNA-based biomarkers distinguish with higher sensitivity and specificity for cancer; therefore, they are more suitable for patient surveillance [16]. Thus, novel molecular markers and genomic classifiers containing molecular information from the RCC (epi)genome, are urgently needed for the careful selection of patients with SRMs eligible for AS and disease aggressiveness prediction.

Changes in DNA methylation occur early during renal carcinogenesis [17], suggesting their suitability as biomarkers for early diagnosis of the disease. Aberrant DNA methylation is a stable modification modulating gene expression. DNA methylation profiles can be associated with the various clinical subgroups of kidney cancer and predict the aggressiveness of the tumor [18]. Moreover, these epigenetic marks are amenable for the detection in liquid biopsies such as urine by conventional and salable PCR methods. Thus, they may be useful as non-invasive biomarkers providing clinicians with an accurate, objective, and rapid tool for the detection and follow-up of renal tumors. Despite efforts made [19,20], no studies are reporting on non-invasive methylated DNA biomarkers for SRMs [21].

Our previous genome-wide methylated DNA analysis allowed us to identify renal cancer-specific methylated DNA biomarkers with a promising potential for non-invasive renal clear cell carcinoma detection and prognosis [22]. In the present study, we analyze the same set of genes, particularly *ZNF677*, *PCDH8*, *FLRT2*, *FBN2*, *TAC1*, and *TFAP2B,* in the urine samples of patients diagnosed with SRMs and treated by AS, aiming at the evaluation of their suitability for the extremely early detection of SRMs and tumor growth prediction during the monitoring of the disease. 

## 2. Results

### 2.1. Tumor Growth Dynamics

Out of 39 SRMs included in the biomarker study, 11 (28.2%) lesions had progressed. Out of them, 9 (81.2%) SRMs had progressed according to the large tumor size on follow-up (>4 cm in maximal diameter) and 2 (18.2%) according to the rapid tumor growth (doubling in volume). All progressed cases had clear cell histology, of which 3 (27.3%) were ISUP grade 1 and 8 (72.7%) ISUP grade 2. Two cases progressed within the first year of AS, 5 cases during the second year of follow-up, and the remaining 4 during the third year or later. Four patients were treated by partial or radical nephrectomy, 4 by ablation, while for the remaining 3 the AS was continued due to the contraindications for active treatment (age and comorbidities). During follow-up, no cases developed metastatic disease; however, one tumor progressed to T3a clinical stage. 

The median maximal tumor diameter at the diagnosis was 2.3 (range 1.0–3.7) cm, while at the last scan it was 2.8 (range 1.2–5.2) cm. Similarly, the median tumor volume at diagnosis was 4.8 (range 0.5–20.9) cm^3^, while at the last scan it was 10.1 (range 0.5–44.1) cm^3^. During the AS period, the average growth rate in maximal tumor diameter of all tumors was 18.1% (0.4 cm), while the volume growth rate was 62.9% (4.5 cm^3^) (Figure 1A,B).

Of importance, during AS the average growth rate in maximal tumor diameter of non-progressing tumors was 1.9% (<0.1 cm), while the same measure of progressing tumors was almost 30-fold larger and reached 52.8% (1.3 cm). Similarly, the volume growth rate of non-progressing tumors was 8.9% (0.6 cm^3^), while of progressing tumors it was 168.5% (13.5 cm^3^). These differences were highly statistically significant (*p* < 0.001). 

While comparing clinical demographic variables, no significant differences between progressing and non-progressing tumors were observed at the initial scan except for tumor histology (Appendix A). A significantly larger size of progressing tumors compared to the non-progressing ones was observed from the third scan, i.e., after ~12 months of AS (*p* < 0.005; Figure 1C,D). 

### 2.2. Diagnostic Potential of the Urinary Biomarkers

DNA methylation of the six genes *TFAP2B, TAC1*, *PCDH8*, *ZNF677*, *FLRT2*, and *FBN2* was analyzed quantitatively in urine sediment samples, collected from the patients diagnosed with SRMs (N = 39) and asymptomatic (N = 92) controls. Methylated DNA levels and/or frequencies of *TFAP2B, TAC1, PCDH8*, *ZNF677,* and *FLRT2* were higher among the patients diagnosed with SRMs as compared to ASC (all *p* < 0.050; Figure 2A,B). 

Assessing the diagnostic potential of each of the genes separately, *PCDH8* (AUC = 0.69; *p* < 0.001) had the highest diagnostic values, with sensitivity and specificity equal to 48.7% and 88.0%, respectively (Figure 2C). The analysis of potential biomarkers in all possible combinations was performed, and the four biomarkers panel, particularly, *ZNF677*, *PCDH8*, *TAC1*, and *FLRT2* was characterized with the best diagnostic potential (AUC = 0.74) reaching 92% sensitivity and 52% specificity (Figure 2D; Appendix A).

### 2.3. Prognostic Value of Urinary Biomarkers

Aberrant methylation of the investigated genes was further analyzed according to the clinical and demographic variables of SRM cases. A significantly higher methylation level of *PCDH8* and *TFAP2B* was observed in females, compared to males (*p* = 0.009 and *p* = 0.042, respectively; Appendix A). In addition, methylated levels of *TFAP2B* and *TAC1* tended to correlate with glomerular filtration rate, while *ZNF677* and *PCDH8* were significantly related to the blood creatinine level (R_s_ = −0.33; *p* = 0.048 and R_s_ = −0.36; *p* = 0.028 respectively; Appendix A).

Most importantly, a significant association of methylated *PCDH8* level to tumor maximal diameter at diagnosis (R_s_ = 0.35; *p* = 0.028) as well as borderline significance to tumor volume (R_s_ = 0.32; *p* = 0.051) at diagnosis was identified (Figure 3A,B). Meanwhile, *TFAP2B* methylation level was negatively associated with the changes in tumor maximal diameter and volume growth rate (Figure 3C,D). Moreover, a comparison of the progressing tumors with non-progressing SRMs revealed a lower level of methylated *TFAP2B* at SRM diagnosis (*p* = 0.016; Appendix A), though no significant changes between different time points during follow-up were detected (Appendix A). 

Assessment of correlations between the methylation levels of described genes and individual tumor growth dynamics revealed some positive associations (Figure 3E–H); however, in most of the cases, the associations were quite stochastic and only partially reflected tumor growth (Appendix A).

## 3. Discussion

The present study of urinary biomarkers shows that SRMs contribute significantly to the amount of DNA detectable in urine; therefore, DNA methylation might be used as an indicator of the formation of small malignant masses in the kidney. In urine samples from the cases with SRMs, significantly higher methylation levels and/or frequencies of *TFAP2B*, *TAC1*, *PCDH8*, *ZNF677*, and *FLRT2* were detected in comparison to controls in support of our previous observations of the diagnostic potential of these DNA methylation biomarkers in RCC [22]. Only the methylation changes in *PCDH8* and *TFAP2B* were related to the amount of tumor mass or the dynamics of tumor growth, thus may potentially supplement imaging data on SRMs’ growth.

AS is a sufficiently safe and reasonable management strategy for the elderly and comorbid patients with SRMs suspicious of clinical T1a RCC. Unfortunately, with or without surgical treatment, there is still a potential for metastatic progression [13]. Optimal management of SRMs should balance the need for treatment, as approximately 25% of all SRMs are benign masses, in order to preserve renal functions as long as possible and to avoid the unnecessary risk of treatment complications [23]. In the present study, core biopsies were conclusive in 91.5% (65 out of 71) cases, identifying 29.2% (19 out of 65) benign masses and 70.8% (46 out of 65) RCCs. Thus, the sensitivity of needle biopsy was slightly lower than previously reported [24,25,26]. Currently, the best way to overcome false negative results is by repeated biopsy [24,26]; however, in addition to the high price, it also provides many difficulties for patients and radiologists, especially in the case of SRMs. Although a previous study established a DNA methylation-based identifier for needle biopsy samples [19], a large amount of variables included in this identifier is of limited clinical use due to the high cost and intricate interpretation. Currently, simple liquid biopsy-based methods are coming to the clinics as an additional tool for radiological follow-up of SRMs. 

In our study, in urine samples from the patients with SRMs, out of five SRM-specific biomarkers, the highest diagnostic power (AUC = 0.69), with 49% sensitivity and 88% specificity, was established for *PCDH8.* These data are in support of our previous findings on the high diagnostic value of this biomarker in RCC [22]. Interestingly, the sensitivities of methylated *TFAP2B* and *TAC1* for RCC SRMs were even higher than that of needle biopsy, which possibly is the result of RCC heterogeneity that is highly overlooked in single biopsy studies [27,28,29]. Moreover, our four biomarkers panel was able to detect >90% of small renal tumors, although the specificity only slightly exceeded 50%. To the best of our knowledge, the present study for the first time analyzed DNA methylation in the urine samples of SRM patients and revealed the possibility to use it as a non-invasive tool for the detection of extremely small (<4 cm in the maximal diameter) malignant renal masses. 

There is an ongoing debate about the treatment of SRMs due to the low rate of progression observed during AS [30]. Until now, serial imagining and tracking the growth rate of SRMs was the only strategy for the surveillance of patients non-eligible for surgical treatment. In this study, among the investigated 39 cases, 28% have progressed demonstrating a considerably higher growth rate (13.5 cm^3^/AS) as compared to non-progressing ones (0.6 cm^3^/AS). However, a significant difference in tumor size among progressing and non-progressing tumors was observed only after about 12 months of AS, but such a tendency was not evident at diagnosis. Meanwhile, the molecular biomarker analysis in the urine samples of SRM RCC patients at diagnosis demonstrated the positive correlation between methylated *PCDH8* level and tumor size at diagnosis, confirming our previous report and its prognostic potential [22]. Some other studies reported tumor suppressive characteristics of *PCDH8*, particularly inhibition of cell proliferation, migration, and induction of apoptosis, and thus this gene is inactivated by promoter methylation in some carcinomas [31,32,33]. Additionally, a significantly higher methylated *TFAP2B* level among non-progressing cases compared to progressing ones at SRM diagnosis was observed as well. Lower methylated *TFAP2B* levels among progressive tumors may be partially explained by its contribution to the activation of the WNT/β-catenin signaling pathway, promoting cell proliferation and RCC progression [34,35,36]. The results suggest the potential of DNA methylation biomarkers to predict SRM progression at disease diagnosis, thus stratifying patients into eligible and ineligible for AS. However, further investigations in pursuance to the application of such prognostic biomarkers in clinical practice, are needed.

Sex differences are known to play a role in renal cancer progression, and outcomes as well and the distinct DNA methylation patterns among males and females may at least partially influence this [37,38]. Although no significant associations between patients’ gender and SRM progression were observed in the present study, significantly higher methylation levels of *PCDH8* and *TFAP2B* were identified among females correlating with the previously observed higher DNA methylation in female representatives among the genes, related to cell proliferation and cancer [38]. However, further thorough investigations are needed to unravel this correlation and rule out the possibility of coincidence. 

Along with the significant impact of the study in the field of AS of SRMs, the investigation has some important shortcomings as well. Firstly, a small number of patients recruited in the study prevented us from performing a more comprehensive statistical analysis. Secondly, next to the asymptomatic control, urine samples from the benign SRMs cases including oncocytomas would be highly desirable to prove the suitability of the investigated biomarkers for non-invasive specific SRM/RCC detection. Thirdly, stochasticity in tumor growth is partially possibly influenced by different imaging methods used (due to the toxicity of CT in some cases), which probably biases the tumor growth measurements that may affect statistical analysis as well. Nonetheless, the innovation of using DNA methylation biomarkers with feasible diagnostic and prognostic value for SRM/RCC and amenable for non-invasive urine-based detection undoubtedly will stimulate further investigations in this field. 

## 4. Materials and Methods

### 4.1. Patient Selection and Biosamples

SRMs patients’ inclusion criteria encompass: (1) patients older than 18 yr; (2) incidental diagnosis at imaging (ultrasonography, CT, MRI) of a solid renal mass <4 cm in maximum diameter (pT1aN0M0); (3) histologically confirmed RCC by percutaneous needle biopsy at diagnosis, all RCC subtypes are eligible for the study; (4) patients inappropriate for active treatment due to advanced age, or co-morbidity, or choosing to avoid active treatment and signed informed consent. Patients’ exclusion criteria: (1) an estimated patients life expectancy of <1 yr; (2) patients on simultaneous systemic therapy for malignancy; (3) patients having a hereditary renal cancer syndrome, and/or had a nondiagnostic (noninformative) or benign biopsy. 

The patients were included between March 2018 and 2022 and monitored at the National Cancer Institute (Vilnius, Lithuania). Approval from the Lithuanian Bioethics Committee was obtained, and all patients gave informed consent for participation. The prospective trial included 125 patients with SRMs, of whom 71 were biopsied. A percutaneous biopsy of the renal mass was performed to histologically confirm the diagnosis of RCC. The histopathologic evaluation of the cores was performed with hematoxylin and eosin (H&E) staining and specific histochemical and immunohistochemical stains when needed to identify specific subtypes of RCC. Among 46 patients with proven SRM RCC, 39 were eligible for the biomarker study (Figure 4). Among these cases, 28 (71.8%) were biopsy-proved clear cell RCC (ccRCC), 7 (17.9%) papillary RCC (pRCC), 3 (7.7%) chromophobe RCC (chRCC), and 1 (2.6%) had mixed ccRCC and pRCC histology. Ten (27.8%) of ccRCC and pRCC tumors had ISUP grade 1, twenty-three (63.9%) grade 2, and for the remaining three (8.3%) cases, the ISUP grade was undetermined. The median patient age at the diagnosis was 78 (IQR 72–81) yr. The median follow-up time was 22 (IQR 12–27) months. Of the patients, 77% were followed for at least 1 yr, 33% for at least 2 yr, and 13% for at least 3 yr. All patient’s clinical and demographic data are provided in Appendix A. 

### 4.2. Imaging Examination and Measurement of Lesions

During the study/follow-up, 165 images were obtained, of which 146 (88.5%) were scanned with computed tomography (CT), 16 (9.7%) were scanned with magnetic resonance imaging (MRI), and 3 (1.8%) were obtained by ultrasound (US). All images were evaluated by expert radiologists.

Serial imaging with CT, MRI, or US has been performed approximately at 3, 6, and 12 months over 3 years. Tumor size was measured by maximal axial diameter. Tumor volume was calculated using the formula for ellipsoid volume calculation: 0.5326 × X × Y × Z. Tumor growth was reported in mm^3^/year (volume growth rate), mm/year referring to the largest 2 diameters at 90 degrees to each other (linear growth rate). Tumor progression was considered in the case of: (1) large size (tumor maximum diameter exceeds 4 cm in at least two measurements within 15 mo), and/or (2) rapid growth (tumor volume doubling over any one-year period) as reported previously [39,40].

### 4.3. Urinary DNA Purification

Serial urine samples were collected from the 39 patients diagnosed with SRMs at different AS time points (165 samples in total): at the beginning of surveillance, and during follow-up. A voided urine sample from 92 asymptomatic volunteers from the previous study [22] was used as asymptomatic control (ASC).

Urine samples (up to 100 mL) were centrifuged at 2000× *g* for 15 min at 4 °C. The supernatant was removed, and urine sediments were washed twice with phosphate-buffered saline. DNA from urine sediments (~2 mL) was released by treating samples for 18 h at 55 °C with 10 μL of proteinase K (Thermo Scientific™, Thermo Fisher Scientific, Wilmington, DE, USA) in 500 μL of lysis buffer, consisting of 175 mM EDTA, 750 mM NaCl, 100 mM Tris-HCl (pH = 8.0), and 1% sodium dodecyl sulfate (all from Carl Roth, Karlsruhe, Germany). DNA was purified following the standard phenol-chloroform protocol as previously reported [41].

### 4.4. Quantitative DNA Methylation Analysis

For DNA methylation analysis of the target genes, up to 400 ng of purified DNA were bisulfite-modified, using EZ DNA Methylation™ Kit (Zymo Research, Irvine, CA, USA) following manufacturer’s instructions, apart from the primary incubation of samples was performed at 42 °C for 15 min. Modified DNA samples were analyzed immediately or stored at ≤−20 °C.

The bisulfite-modified DNA was used as a template for the quantitative methylation-specific polymerase chain reaction (QMSP). To normalize DNA input, *ACTB* was included in each assay as the endogenous control gene [42]. The QMSP primers and hydrolysis probes, specific for the methylated DNA of the genes *PCDH8*, *TFAP2B*, *TAC1, ZNF677*, *FLRT2,* and *FBN2* were designed as described previously [22] and ordered from Metabion (Martinsried, Germany). The rection for each set of primers was performed in triplicate in the separate wells as described previously [22].

### 4.5. Statistical Analysis

Statistical analysis was performed using MedCalc^®^ v14.0 software (MedCalc Software, Ostend, Belgium), STATISTICA™ v8.0 (StatSoft, Tulsa, OK, USA), and GraphPad Prism v8.0.1 (GraphPad Software, Inc., San Diego, CA, USA). For quantitative variables, considering the non-normal (skewed) distribution, a nonparametric Mann–Whitney U test was applied for the comparison between the two groups. To test the associations between two continuous variables, Spearman’s (R_S_) rank correlation coefficients were calculated. To evaluate the diagnostic value of the biomarkers to distinguish SRMs and ASC, receiver operating characteristic (ROC) curve analysis and estimation of the area under the curve (AUC) were performed. The diagnostic test’s performance parameters—specificity and sensitivity—were obtained based on the Youden index. Logistic regression analysis was applied for various combinations of biomarkers. Differences and associations were assumed statistically significant at *p* < 0.050.

## 5. Conclusions

To sum up, the RCC-specific aberrant methylation of *TFAP2B*, *TAC1*, *PCDH8*, *ZNF677*, and *FLRT2* is detectable in urine from patients with SRM and is potentially suitable for non-invasive early detection and follow-up of the disease. Moreover, *PCDH8* and *TFAP2B* methylation show some potential as prognostic biomarkers for SRM patients’ active surveillance, although further thorough investigations are needed. 

## Figures and Tables

**Figure 1 ijms-23-16110-f001:**
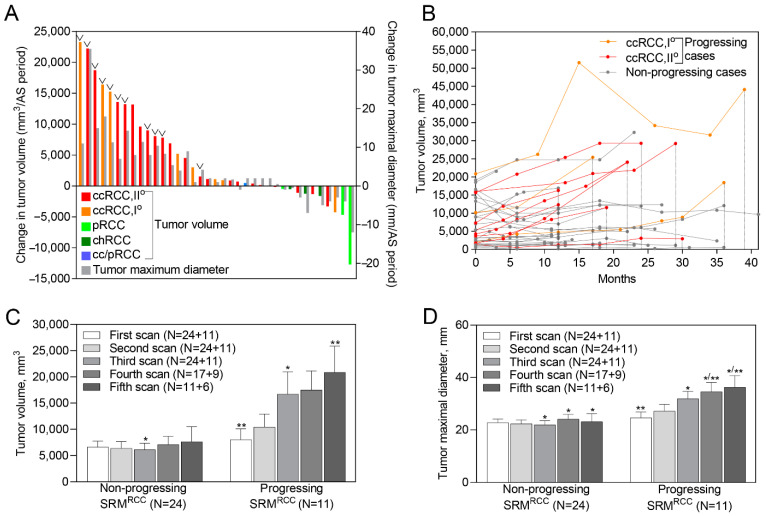
Growth kinetics of biopsy confirmed SRM RCC. (**A**) Waterfall plots of tumor growth rate during all AS periods. (**B**) Tumor growth kinetics at different time points of the AS period. (**C**,**D**) Average tumor size at different scans. Whiskers indicate the standard error of the mean. ˅ indicates progressing cases; * indicates significant differences between the groups (Non-progressing vs. Progressing); ** indicates significant differences within the group (between the first scan and the subsequent scans). SRM—small renal masses; (cc)RCC—(clear cell) Renal cell carcinoma; AS—active surveillance.

**Figure 2 ijms-23-16110-f002:**
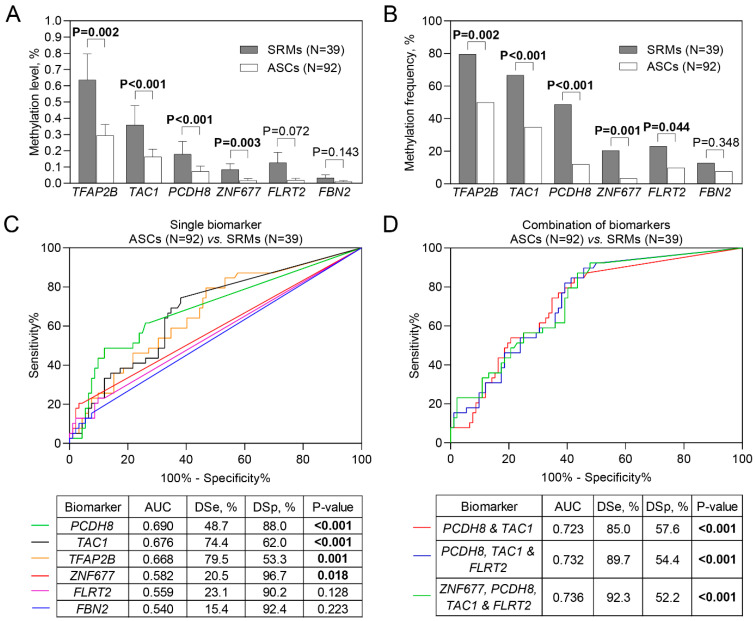
Diagnostic characteristics of the investigated biomarkers. (**A**) Methylation levels of *TFAP2B*, *TAC1, PCDH8*, *ZNF677*, *FLRT2,* and *FBN2* in the urine samples from patients diagnosed with small renal masses (SRMs; pT1a; ≤4 cm) and asymptomatic controls (ASC). (**B**) Frequencies of methylated *TFAP2B*, *TAC1*, *PCDH8*, *ZNF677*, *FLRT2,* and *FBN2* among the patients diagnosed with SRMs and ASCs. (**C**) ROC curve analysis for single biomarkers. (**D**) The combination of biomarkers in distinguishing patients with SRMs and ASC. The bars represent the average DNA methylation levels with a standard error of the mean. ROC—receiver operating characteristic; AUC—area under the curve; DSe—diagnostic sensitivity; DSp—diagnostic specificity. Significant *p* values are in bold.

**Figure 3 ijms-23-16110-f003:**
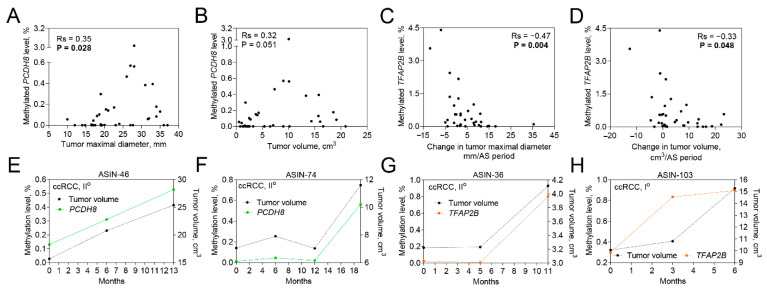
The association of investigated biomarkers with SRM size and growth rate. (**A**,**B**) Correlation between methylated *PCDH8* level and SRM size at diagnosis. (**C**,**D**) Correlation between methylated *TFAP2B* level and change in tumor size during AS period. (**E**–**H**) Associations between individual tumor growth dynamics and methylated *PCDH8* and *TFAP2B* levels. Rs—Spearman correlation coefficient; AS—active surveillance; ccRCC—clear cell renal cell carcinoma. ASIN depicts the patient’s ID. Significant *p*-values are in bold.

**Figure 4 ijms-23-16110-f004:**
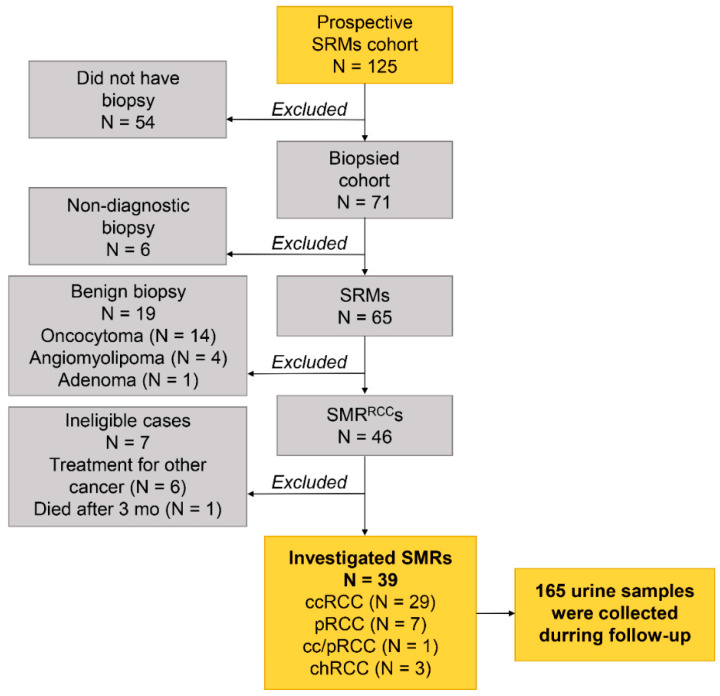
Consort diagram of the patients with SRMs managed by active surveillance. SRM—small renal mass; RCC—renal cell carcinoma; ccRCC—clear cell RCC; pRCC—papillary RCC; chRCC—chromophobe RCC; mo—month.

## Data Availability

All data supporting the results reported in the article is available from the corresponding author upon a reasonable request.

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
