# Peer review of "Urine Molecular Biomarkers for Detection and Follow-Up of Small Renal Masses"

_ijms, 2022, doi:10.3390/ijms232416110_

Round 1

Reviewer 1 Report

Please 

1-mention the the main findings of your previous study in a separate paragraph and explain the differences in the methodology and design and how this study can further your cause, (reference 20)

2-please separete the methods that was used in the previous study and mention the necessary parts for the current study

3-Please justify using non-parametric tests instead of parametric tests in the statistical analysis

4-Please explain more about the p value within the group in Figure 2

5-Please arrange tables in Figure 3 based on the AUC (increasing/decreasing)

4-What kind of correlation was used in fig 4? Pearson's correlation is a parametric test

5-first paragraph in discussion should be dedicated to the main findings of the current study and not the previous one (ref 20)

Author Response

Reviewer 1

Comments and Suggestions for Authors

  1. Mention the main findings of your previous study in a separate paragraph and explain the differences in the methodology and design and how this study can further your cause, (reference 20)

Response: Thanks to the reviewer for pointing this out. The main findings of our previous work have been included in the revised version of the manuscript, emphasizing the distinctions from the present work.  The main difference between the previous and the present work is that previously we have performed genome-wide methylated DNA analysis in the cancerous and non-cancerous renal tissue samples and investigated the suitability of the selected genes ZNF677, PCDH8, FLRT2, FBN2, TAC1, and TFAP2B for non-invasive urine-based detection of renal clear cell carcinoma by performing discovery and validation analysis in the two independent cohorts. These cohorts consisted of patients, diagnosed with 57 mm and 49 mm size tumors on average respectively and about half (54% and 46% respectively) were pT3-pT4 stage tumors. Meanwhile, in the present study, we sought to evaluate the suitability of these biomarkers for extremely early detection of small mass renal tumors (average tumor size 23 mm; pT1a stage) and assess their applicability for tumor growth prediction during the active surveillance of these patients.

  1. Please separate the methods that were used in the previous study and mention the necessary parts for the current study

Response: The description of Quantitative DNA methylation analysis was abbreviated by providing a link to the previous study (reference 20).

  1. Please justify using non-parametric tests instead of parametric tests in the statistical analysis

Response: The clarification of the usage of the non-parametric tests was included in the “Statistical analysis” section of the manuscript. All quantitative variables were tested for normality by the Shapiro-Wilk test and since the data significantly deviate from a normal distribution (P < 0.001), nonparametric tests were applied for the following statistical analysis.

  1. Please explain more about the p-value within the group in Figure 2

Response: Addressing the reviewer’s concern, in the revised version of the manuscript, we have explained in slightly more detail the meaning of “significant differences within the group”. In Figures 2 C and D, the double asterisk (**) depict significant differences in tumor size between the first scan and the subsequent scans within the group (when only non-progressing or only progressing cases are compared), while the single asterisk (*) depicts significant differences among the corresponding scans between the groups (Non-progressing vs. Progressing).

  1. Please arrange tables in Figure 3 based on the AUC (increasing/decreasing)

Response: Thanks to the reviewer for the suggestion.  In the revised version of the manuscript, we have arranged the provided genes according to the decreasing AUC values in Figure 3C, while in the D part of the same figure, the combinations of the biomarkers are arranged according to the increasing number of genes in the panel.

  1. What kind of correlation was used in fig 4? Pearson's correlation is a parametric test.

Response: Because of non-normal data distribution, Spearman's rank (RS) correlation coefficients were calculated to test the associations between two continuous variables.

  1. First paragraph in discussion should be dedicated to the main findings of the current study and not the previous one (ref 20)

Response: We agree with the reviewer on this point. The main findings provided in the first paragraph of the discussion are based on the results obtained in the present study. The previous work (reference 20) is mentioned only to advert the repeatability of the higher methylation levels of TFAP2B, TAC1, PCDH8, ZNF677, and FLRT2 in the urine samples from patients, diagnosed with renal cell carcinoma.

Reviewer 2 Report

Overview and general comments:

The current study attempts to validate new DNA methylation biomarkers for non-invasive SRM detection and prognosis. The levels of methylated TFAP2B, TAC1, PCDH8, ZNF677, FLRT2, and FBN2 were determined using a quantitative methylation-specific polymerase chain reaction in 165 serial urine samples collected prospectively from 39 patients diagnosed with SRM, specifically renal cell carcinoma (RCC), before and during the active surveillance.

I found the paper to be overall very well written and I felt confident that the authors performed careful literature analysis and research data interpretation. I recommend that a minor revision of the manuscript is warranted. I explain my concerns in more detail below and I would ask that the authors will correct them.

Minor comments:

Line 13: “ the novel DNA methylation biomarkers” the novel word is not appropriated since the methylation of the specific genes indicated in this paragraph are not new and novel DNA methylation biomarkers since they were recently published in J Cancer Res Clin Oncol. 2022 Feb;148(2):361-387 375. doi: 10.1007/s00432-021-03837-7, ref 20 in the manuscript. We usually use the word novel when we present something different from anything seen or known before.

Line 14: please mention the term genes before introducing their codes, some less experts might not know that they are genes

Line 121: 8 h should be 8 hrs.

Line 143: no hyphen is not necessary between 20 and μL

Line222: Some statements should be correlated with what is known already in the literature, e.g. “significantly higher methylation level of PCDH8 and TFAP2B was observed in females, compared to males” which is already studied for other gens and other fluids. Please indicate some more references and correlation. See “Characterising sex differences of autosomal DNA methylation in whole blood using the Illumina EPIC array”, doi: 10.1186/s13148-022-01279-7.

Line 228:  the mention "indicating the possible contribution of these genes to the kidney functions” might be supported by some references, even related to some proteins derived from the respective genes ZNF677 and PCDH8

Please also check the plagiarism report and modify as much as possible the text with a high percentage of similarity, even in the methods section.

Author Response

Reviewer 2

Overview and general comments:

The current study attempts to validate new DNA methylation biomarkers for non-invasive SRM detection and prognosis. The levels of methylated TFAP2B, TAC1, PCDH8, ZNF677, FLRT2, and FBN2 were determined using a quantitative methylation-specific polymerase chain reaction in 165 serial urine samples collected prospectively from 39 patients diagnosed with SRM, specifically renal cell carcinoma (RCC), before and during the active surveillance.

I found the paper to be overall very well written and I felt confident that the authors performed careful literature analysis and research data interpretation. I recommend that a minor revision of the manuscript is warranted. I explain my concerns in more detail below and I would ask that the authors will correct them.

Response: We thank the Reviewer for their careful reading of the manuscript and appreciation.

Minor comments:

  1. Line 13: “the novel DNA methylation biomarkers” the novel word is not appropriated since the methylation of the specific genes indicated in this paragraph are not new and novel DNA methylation biomarkers since they were recently published in J Cancer Res Clin Oncol. 2022 Feb;148(2):361-387 375. doi: 10.1007/s00432-021-03837-7, ref 20 in the manuscript. We usually use the word novel when we present something different from anything seen or known before.

Response: Many thanks to the reviewer for their constructive remarks. The word “novel” has been omitted from line 13.

  1. Line 14: please mention the term genes before introducing their codes, some less experts might not know that they are genes

Response: According to the reviewer’s concern, for clarity, the word “genes” has been inserted in line 14 before the abbreviated names of the investigated biomarkers.

  1. Line 121: 8 h should be 8 hrs.

Response: Thanks to the reviewer for noticing, the line has been corrected.

  1. Line 143: no hyphen is not necessary between 20 and μL

Response: In accordance with Reviewer #1 concern, the content included in the corresponding line has been removed from the manuscript.

  1. Line 222: Some statements should be correlated with what is known already in the literature, e.g. “significantly higher methylation level of PCDH8 and TFAP2B was observed in females, compared to males” which is already studied for other gens and other fluids. Please indicate some more references and correlation. See “Characterising sex differences of autosomal DNA methylation in whole blood using the Illumina EPIC array”, doi: 1186/s13148-022-01279-7.

Response: Thanks to the reviewer for the suggestion to try to puzzle out these associations, though this is beyond the scope of the present study. Some correlations have been provided in the discussion section of the revised manuscript in accordance with the recommended article.

  1. Line 228:  the mention "indicating the possible contribution of these genes to the kidney functions” might be supported by some references, even related to some proteins derived from the respective genes ZNF677 and PCDH8

Response: The main known functions of ZNF677 and PCDH8 were described in our previous work (reference 20) and to avoid repetition, are not exhaustively discussed in the present study. ZNF677 (zinc finger protein 677), encodes the transcription factor belonging to the zinc finger protein family. ZNF677 may function as a tumor suppressor regulating the transcription of many genes and its overexpression in cancer cells was related to the inhibition of cell proliferation, migration, invasion, tumorigenic potential, and induction of cell-cycle arrest and apoptosis; while down-regulation has an opposite

effect [1,2]. More specifically, it is observed that ZNF677 induces G0–G1 phase arrest, inhibits Akt phosphorylation, and activates p53 signaling, partially through transcriptionally repressing its targets, e.g., CDKN3 [2]. PCDH8 (Protocadherin-8) encodes a transmembrane protein belonging to the protocadherins, participating in cell adhesion, proliferation, differentiation, and migration processes [3]. There are sole functional studies on ZNF677 in the case of ccRCC, showing its role in the inactivation of the PI3K/AKT signaling pathway and inhibition of epithelial-mesenchymal transition [4], however, no correlations with renal functions, related to creatinine filtration can be made. On the other hand, blood creatinine level itself has some important limitations for the determination of renal functions [5], thus further thorough investigations are needed to unravel the observed negative correlation between ZNF677 and PCDH8 methylation and creatinine level.

References:

[1] Heller G, Altenberger C, Schmid B, Marhold M, Tomasich E, Ziegler B,  Müllauer L, Minichsdorfer C, Lang G, End-Pfützenreuter A, Döme B, Arns BM, Fong KM, Wright CM, Yang IA, Klepetko W, Zielinski CC, Zöchbauer-Müller S. DNA methylation transcriptionally regulates the putative tumor cell growth suppressor ZNF677 in non-small cell lung cancers. Oncotarget. 2015;6(1):394-408.

[2] Li Y, Yang Q, Guan H, Shi B, Ji M, Hou P. ZNF677 Suppresses Akt Phosphorylation and Tumorigenesis in Thyroid Cancer. Cancer Res. 2018;78(18):5216-5228

[3] Li Y, Yang Q, Guan H, Shi B, Ji M, Hou P. ZNF677 Suppresses Akt  Phosphorylation and Tumorigenesis in Thyroid Cancer. Cancer Res. 2018;78(18):5216-5228

[4] Liang W, Chen S, Yang G, Feng J, Ling Q, Wu B, Yan H, Cheng J. Overexpression of zinc-finger protein 677 inhibits proliferation and invasion by and induces apoptosis in clear cell renal cell carcinoma. Bioengineered. 2022;13(3):5292-5304.

[5] Thompson EL. and Joy MS. Endogenous markers of kidney function and renal drug clearance processes of filtration, secretion, and reabsorption. Curr. Opin. Toxicol. 2022. 31:100344

  1. Please also check the plagiarism report and modify as much as possible the text with a high percentage of similarity, even in the methods section.

Response: To avoid plagiarism as much as possible, the text with a high percentage of similarity with the previous works (mostly of our group) has been modified in the revised version of the manuscript.

Reviewer 3 Report

Well written manuscript. Minor comments

- Authors briefly mentioned liquid biopsy. Do they have any experience in studying the concordance of the detection rates of molecular alterations between the urine and serum samples?

-There is a proposed model for the prognostication of SRMs.  Please discuss the same ( refer to https://onlinelibrary.wiley.com/doi/full/10.1002/ijc.32650)

-Why was oncocytoma excluded? Consider discussing this, at least in the discussion. I see other authors including the same, so curious to know about this, at least in the discussion, if there is any relevant information on this subtype (https://academic.oup.com/jnci/article/91/23/2028/2606696)

Author Response

Reviewer 3

Comments and Suggestions for Authors

Well-written manuscript.

Response: We thank the Reviewer for their appreciation.

Minor comments

  1. Authors briefly mentioned liquid biopsy. Do they have any experience in studying the concordance of the detection rates of molecular alterations between the urine and serum samples?

Response: Unfortunately, we have no experience in studying the concordance of the detection rate of methylated DNA alterations between the urine and serum samples in the case of renal cancer. However, our previous experience with miRNA studies in the urine and blood of prostate cancer patients revealed urine as a more reliable source for cancer-specific nucleic acid studies in liquid biopsy. Similarly, the study conducted by Hoque et al. observed considerably lower diagnostic sensitivity of methylated DNA biomarkers for renal cancer in the serum-based analysis when compared to the urine-based [1]. This may be partially explained by the fact, that the half-life of circulating DNA in the bloodstream is between 16 min and 2 hours [2] whereas in urine up to 5 hours [3], thus is more concentrated in the latter. In addition, considering the completely non-invasive nature of the obtainment of urine samples, this makes urine a more convenient source for the search for molecular biomarkers and application in routine clinical practice.

References:

[1] Hoque, M.O.; Begum, S.; Topaloglu, O.; Jeronimo, C.; Mambo, E.; Westra, W.H.; Califano, J.A.; Sidransky, D. Quantitative detection of promoter hypermethylation of multiple genes in the tumor, urine, and serum DNA of patients with renal cancer. Cancer Res. 2004, 64, 5511–5517.

[2] Diehl F, Schmidt K, Choti MA, Romans K, Goodman S, Li M, Thornton K, Agrawal N, Sokoll L, Szabo SA, Kinzler KW, Vogelstein B, Diaz LA Jr. Circulating mutant DNA to assess tumor dynamics. Nat Med. 2008 Sep;14(9):985-90.

[3] Cheng THT, Jiang P, Tam JCW, Sun X, Lee WS, Yu SCY, Teoh JYC, Chiu PKF, Ng CF, Chow KM, Szeto CC, Chan KCA, Chiu RWK, Lo YMD. Genomewide bisulfite sequencing reveals the origin and time-dependent fragmentation of urinary cfDNA. Clin Biochem. 2017 Jun;50(9):496-501.

  1. There is a proposed model for the prognostication of SRMs.  Please discuss the same (refer to https://onlinelibrary.wiley.com/doi/full/10.1002/ijc.32650)

Response: Thank the Reviewer for this recommendation. The provided article has been mentioned in the introduction of the revised manuscript. As a biomarker for cancer, circulating tumor DNA have several advantages over protein analytes. Firstly, circulating tumor DNA possesses more specificity for cancer than protein biomarkers [1]. Secondly, because of the short half-life of circulating DNA compared to proteins (several days) [2], DNA should be more sensitive for identifying rapid changes in tumor burden and in monitoring [2]. Third but not least, DNA is much more stable than protein and can be detected by inexpensive standard PCR methods, thus is more compatible with the application for convenient non-invasive urine-based molecular tests.

References:

[1] Duffy MJ, Crown J. Circulating Tumor DNA as a Biomarker for Monitoring Patients with Solid Cancers: Comparison with Standard Protein Biomarkers. Clin Chem. 2022;68(11):1381-1390.

[2] Heitzer E, Haque IS, Roberts CES, Speicher MR. Current and future perspectives of liquid biopsies in genomics-driven oncology. Nat Rev Genet. 2019;20(2):71-88.

  1. Why was oncocytoma excluded? Consider discussing this, at least in the discussion. I see other authors including the same, so curious to know about this, at least in the discussion, if there is any relevant information on this subtype (https://academic.oup.com/jnci/article/91/23/2028/2606696)

Response: According to the European Association of Urology, Renal cell carcinoma [1], and American Urological Association (AUA) guidelines [2], active surveillance is not recommended for benign renal tumors and is significantly different from the surveillance algorithm for malignant renal tumors. Not only the phasing, but methods of visual examination are different. Based on this we excluded them. However, in the discussion part of the revised manuscript, we mentioned that as possible drawbacks of the present study. Notwithstanding, oncocytomas are sophisticated benign renal masses, some of which have never progressed to malignant renal masses at all [3]. Thus, we agree with the reviewer, that it would be fascinating to analyze these cases as well, but it needs separate thorough investigations.

References:

[1] Ljungberg B, Albiges L, Abu-Ghanem Y, Bedke J, Capitanio U, Dabestani S, Fernández-Pello S, Giles RH, Hofmann F, Hora M, Klatte T, Kuusk T, Lam TB, Marconi L, Powles T, Tahbaz R, Volpe A, Bex A. European Association of Urology Guidelines on Renal Cell Carcinoma: The 2022 Update. Eur Urol. 2022 Oct;82(4):399-410. 

[2] Campell SC, Clark PE, Chang SS et al: Renal Mass and Localized Renal Cancer: Evaluation, Management, and Follow-Up: AUA Guideline Part I. J Urol 2021; 206: 199.

[3] Joshi S, Tolkunov D, Aviv H, Hakimi AA, Yao M, Hsieh JJ, Ganesan S, Chan CS, White E. The Genomic Landscape of Renal Oncocytoma Identifies a Metabolic Barrier to Tumorigenesis. Cell Rep. 2015;13(9):1895-908.

Round 2

Reviewer 1 Report

Thank you very much

no further comments